


# Hypsometric amplification and routing moderation of Greenland ice sheet meltwater release

Dirk van As[1], Andreas Bech Mikkelsen[2], Morten Holtegaard Nielsen[3], Jason E. Box[1], Lillemor

Claesson Liljedahl[4], Katrin Lindbäck[5], Lincoln Pitcher[6] and Bent Hasholt[2]

[1]Geological Survey of Denmark and Greenland, Øster Voldgade 10, 1350 Copenhagen, Denmark

[2]Department of Geosciences and Natural Resource Management, University of Copenhagen, Øster Voldgade 10, 1350 Copenhagen, Denmark

[3]Marine Science & Consulting, Peder Lykkes Vej 8, 4. th, 2300 Copenhagen, Denmark

[4]Svensk Kärnbränslehantering AB, Research and Safety Assessment, Box 250, SE-101 24 Stockholm, Sweden

[5]Norwegian Polar Institute, Framsentret, Postboks 6606, Langnes, 9296 Tromsø, Norway

[6]Department of Geography, University of California – Los Angeles, Los Angeles, California, USA 90095

*Correspondence to:* Dirk van As (dva@geus.dk)

**Abstract.** Concurrent ice sheet surface runoff and proglacial discharge monitoring are essential for understanding Greenland ice sheet meltwater release. We use an updated, well-constrained river discharge time series from Watson River in southwest Greenland, with an accurate, observation-based ice sheet surface mass balance model of the ca. 12,000 km$^2$ ice sheet area

feeding the river. For the 2006-2015 decade, we find that the large, factor of three range in interannual variability is for ca. 56% caused by hypsometric amplification through ice sheet area increase with elevation. A good match between river discharge and ice sheet surface meltwater production is found after introducing elevation-dependent transit delays that moderate diurnal variability in meltwater release by a factor of 10-20. The routing lag time increases with ice sheet elevation and attains values in excess of one week for the upper reaches of the runoff area at ca. 1800 m above sea level. These multi-

day routing delays yield that the highest proglacial discharge levels, and thus overbank flooding events, are more likely to occur after multi-day melt episodes. Finally, we conclude that there is little evidence of meltwater storage in or release from the en- and subglacial environments based on the unprecedented good match between ice sheet runoff and proglacial discharge.



## 1 Introduction

The majority of recent Greenland ice sheet mass loss can be attributed to increases in surface melting (Enderlin et al., 2014). Several methods exist to determine ice sheet mass balance, including remotely-sensed changes in gravity (e.g. Velicogna 2009). Yet gravimetric estimates alone cannot distinguish between changes in surface mass balance (SMB) and dynamic mass loss through solid ice discharge. Conventional tools for determining Greenland-wide SMB are atmospheric and land surface models forced by atmospheric reanalysis data (Fettweis, 2007; Ettema et al., 2009; Langen et al., 2015). Regionally, SMB models can be driven by in-situ weather station observations (Van As et al., 2012). These models resolve the components of the surface energy and mass budgets, and therefore allow evaluation of physical processes impacting SMB, such as snow accumulation, surface melting, refreezing in snow and firn, and sublimation.

Another method for studying regional SMB is by monitoring proglacial river discharge comprised of melt- and rainwater exiting the ice sheet without getting retained in supra-, en, and subglacial environments and not otherwise lost to evaporation or groundwater storage before being gauged. Along the land-terminating sectors of the Greenland ice sheet, hundreds of proglacial rivers transport sizeable meltwater fluxes to surrounding seas. Studies of proglacial river discharge were initiated during the 1970s and '80s in response to the growing interest in Greenland hydropower (ATV, 1981), but in recent years, attention turned to understanding ice sheet surface mass balance and hydrology (Mernild and Hasholt, 2009; Bartholomew et al., 2011; Rennermalm et al., 2012; Banwell et al., 2013; Van As et al., 2014; Overeem et al., 2015; Smith et al., 2015). Yet still relatively little is known about Greenland ice sheet freshwater discharge in terms of delays imposed by limitations in hydrological efficiency, or because of supra-, en- and subglacial storage.

The complexity of freshwater discharge delay partly stems from the various supra-, en- and subglacial transit processes that are involved, most notably subglacial drainage channel opening and closure in response to meltwater supply (Fountain and Walder, 1998). Studies quantifying the freshwater transit time between ice sheet surface generation and release at the margin are scarce. In terms of surface routing before entering a moulin, Zuo and Oerlemans (1996) hypothesized that drainage of surface water takes between one and 26 days depending on surface slope. For several moulin sites, Chandler et al. (2013) determined en- and subglacial flow velocities to be in the $0.2 - 1.6$ km h$^{-1}$ range, resulting in routing delays from hours to days, depending on the distance travelled and the efficiency of the subglacial drainage system (Meierbachtol et al., 2013). Subglacial lakes are known to exist and drain in Greenland (e.g. Palmer et al., 2015), but we have no precise estimate of the contribution of such retention to regional ice sheet discharge on intra- and interannual time scales. Efforts have been made using the input-output method, in which the difference between ice sheet surface runoff and proglacial discharge provide an estimate of retention. For instance, Rennermalm et al. (2013) calculate retention of up to half of the water running off the surface of the ice sheet.



In this study, we apply similar methods to determine the routing delays as a function of meltwater origin elevation to quantify ice sheet hydraulic transmission efficiency. Our study area is located in southwest Greenland (Fig. 1). We make use of time series of supraglacial meltwater production as calculated by an in-situ, observation-driven SMB model, and observed proglacial river discharge. Both time series stem from earlier research, but have been revised to increase absolute accuracy. Furthermore, by studying the large "Kangerlussuaq" ice sheet catchment, the potential impact of errors in catchment delineation is reduced. In interpreting the discharge time series, special attention is given to how ice sheet discharge responds to climate variability. Previous studies have identified the role of the area-elevation distribution, a.k.a. *hypsometry* (Van As et al., 2012; McGrath et al., 2013; Mikkelsen et al., 2016), but did not quantify as we do here. Hypsometry is expected to be an amplifier of meltwater runoff from the Greenland ice sheet in a warming climate.

## 2 Methods

### 2.1 River discharge

The Watson River drains an ice sheet area of ca. 12,000 km$^2$ (Lindbäck et al., 2015) through confluent branches into the "Kangerlussuaq" fjord (Fig. 1; Nielsen et al., 2010). River stage has been monitored near the bridge in Kangerlussuaq, west Greenland since 2006 (Fig. 2) (Hasholt et al., 2013). The location of pressure transducers 140-150 m upstream of the bridge in between the two main river channels was chosen to be as close to the river bottom as possible to capture low water levels, and sufficiently far upstream to avoid influence of the drop in water level across the stable control section of the river. Water pressure measurements are adjusted for barometric pressure before converting into hourly averages of water stage. River discharge measurements taken over a range of water stages are required to construct a rating curve for converting stage into discharge (e.g. Rennermalm et al., 2012). We use three different methods relating stage and discharge (Hasholt et al., 2013):

1) At very low water levels in spring we wade across the river measuring water velocity with an OTT C2 propeller current meter . The depth- and width-integrated water flux has an uncertainty <10% when the full width of the river is surveyed.

2) From the bridge, we release a tethered float to estimate the depth- and width-averaged flow velocity based on travel time and distance. From 2007-2015 ca. 200 float measurements were obtained from which we select the 13 measurements for which we can constrain the cross sectional area. Only few measurements pass this criterion because of depth variations in channel 1 (Fig. 2) in excess of 2 m, due to erosion and deposition of bed load (sediment and gravel) (Hasholt et al., 2013). One high-discharge 2011 measurement is available in which a boat was used as a float in a wider, echo-sounded section of the river. Hasholt et al. (2013) estimate the float method to be accurate within 15%.





3) In 2012, 74 river crossings were done to perform Acoustic Doppler current profiler (ADCP) measurements by means of a 600 kHz WorkHorse Monitor ADCP from Teledyne RDI, mounted downward-looking over the side of a boat. Concurrently geographic position of the boat is determined using a hand-held GPS receiver. For each river crossing the discharge is calculated by integrating the true flow velocities from bank to bank taking into account the direction and the length of the survey path. Where the flow velocities could not be measured, primarily at the surface, bottom and banks, the flow velocities are assumed to be equal to the mean flow velocity at the location at hand. The combined uncertainty of each depth- and width-integrated river discharge value is estimated to be at most 5% + 100 m$^3$ s$^{-1}$.

The three methods provide 91 near-instantaneous river discharge values to construct a stage-discharge relation. In Fig. 3 we plot discharge versus stage measured at the pressure transducer site. Least-squares fitting of a power function reveals the stage-discharge relation:

$$D = 7.51 \cdot H^{2.340} \qquad (1)$$

with river discharge $D$ in m$^3$ s$^{-1}$ and stage $H$ in m units. The exponent value has theoretical foundation in falling within the 2-3 range that is common in hydraulic and fluvial morphology (Hershey, 1999). We find a fit correlation of r = 0.994 and a root mean square difference of 72 m$^3$ s$^{-1}$. An uncertainty of 8% encompasses all ADCP measurement uncertainties, but we use a conservative uncertainty value of 15% for converting stage into discharge. The best fit suggests that the bottom of the stable bedrock control cross section is 1.97 m below the lowest pressure transducer. Here no sediment is observed to accumulate towards the end of the melt season. In autumn, ice does accumulate in the control cross section that allows for a stable stage-discharge relation, but melts shortly after the river starts to flow in spring.

ADCP measurements were largely unavailable to previous studies of Watson River discharge (Mernild and Hasholt, 2009; Hasholt et al., 2013). The availability of ADCP data considerably changes the stage-discharge relation (Fig. 3), effectively doubling discharge from Hasholt et al. (2013), and quadrupling the values by Mernild and Hasholt (2009). The latter two differ because of a revision in the cross-sectional area and larger availability of float-derived measurements at high stage. The further increase that our rating curve yields, we speculate to be due to remaining uncertainties in cross-sectional area and/or uncertainties in deriving the channel average velocity from surface float measurements in the rapid, supercritical flow through the irregular and seasonally heavily sedimented channel 1 (Fig. 2). Neither error source affects ADCP measurements taken elsewhere in the river.

An independent method to validate river discharge values is presented by the occasional freshwater outbursts (or jökulhlaups) of an ice-dammed lake at the ice sheet margin within the Kangerlussuaq catchment (Russell et al., 2011), causing distinct peaks in river discharge (Fig. 4) (Mernild and Hasholt, 2009; Mikkelsen et al., 2013). Lake volume changes during these outbursts can be compared with the time-integrated spike in river discharge at the Kangerlussuaq bridge, after





adjustment for background flow. Russell et al. (2011) determined ice-dammed lake drainage volumes to be $39.1\pm0.8\cdot10^6$ m$^3$ in 2007 and $12.9\pm0.3\cdot10^6$ m$^3$ in 2008. Discharge values calculated from the previous stage-discharge relations underestimated these jökulhlaup volumes well beyond uncertainty. Our updated stage-discharge relation provides jökulhlaup volume estimates close to those by Russell et al. (2011), totalling $43.1\pm8.6\cdot10^6$ m$^3$ in 2007 and $9.4\pm1.9\cdot10^6$ m$^3$ in 2008. The
2008 event remains underestimated using the updated stage-discharge relation, suggesting that a further increase in the river discharge calculation at low river stage may be appropriate.

## 2.2 Gap filling: Temperature-based discharge

Due to the risk of frost damage, the pressure transducers recording stage cannot remain installed through winter. Consequently, early- and late-season periods exist during which water stage is not recorded. This includes instances during which water stage falls below the level of the pressure transducers. A significant data gap also exists in the 2006 data record when water stage exceeded the pressure transducer's measurement range.

To estimate river discharge during data gaps, we use hourly air temperature data collected in Kangerlussuaq (Cappelen, 2016) as an ice sheet melt proxy. We find in plotting all available river discharge data versus air temperature that a power law approximates their relation (Fig. 4). To roughly account for transit time between the ice sheet surface and the river monitoring site, we apply a ten-day smoothing and five-day delay to the temperature data. We distinguish between the first half of the year (up to and including June) and the second half, because of differences in winter-accumulated snow on tundra
and ice sheet, impacting meltwater retention and retardation.

We find that river discharge (in m$^3$ s$^{-1}$) can be approximated from Kangerlussuaq air temperature $T_0$ (in °C) by:

$$D_T = F_T \cdot T_0^{3.4} \tag{2}$$

Figure 4 illustrates that the temperature response factor $F_T$ equals 0.31 during the peak and late melt season (July and after,
when the ice sheet ablation area has little to no snow cover). $F_T$ is smaller (0.17) during the first half of the year. We set a large, conservatively chosen 70% uncertainty range (Fig. 4) to encompass most discharge measurements except during ice-dammed lake drainages. We return to these equations when quantifying ice sheet hypsometric amplification of meltwater runoff.

In gap filling, $D_T$ makes up 69% of the 2006 river discharge total, but adds up to just 1.2% of the discharge spanning 2007-2015. Hasholt et al. (2013) found higher values of 4-11% of annual river discharge that is not registered by the pressure transducers during low flow conditions in 2007-2010. The fact that we find a smaller contribution during data gaps can largely be attributed to the revised stage-discharge relation.



## 2.3 Surface mass balance modelling

Supraglacial runoff is calculated by an improved version of the SMB model by Van As et al. (2012) and has been used

successfully for different glacier settings around the globe (e.g. Van As et al., 2005; Van den Broeke et al., 2008). The model has the advantage of being forced with local observations, as opposed to e.g. regional climate models that are constrained at remote boundaries. The model interpolates/extrapolates meteorological and radiation data from three automatic weather stations placed at different ice sheet elevations into 100 m elevation bins and calculates the surface energy and mass components in each bin, ranging from the margin to 2000 m above sea level (ASL), above the surface runoff limit. For every

time step, the model iteratively solves the surface energy balance for the surface temperature. If surface temperature is limited by the melting point, the surplus energy is used for melting of snow or ice.

In calculating turbulent heat fluxes, we set aerodynamic surface roughness for momentum to commonly accepted and observed values of 0.02 and 1 mm for snow and ice, respectively (Van As et al., 2005; Smeets and Van den Broeke, 2008).

We adopt a snow density value of 500 kg m$^{-3}$ after Van den Broeke et al. (2008). Snow and firn densify and gain heat in the model through the refreezing of meltwater that percolates from the surface, provided cold content is available (Charalampidis et al., 2015), and that no "impenetrable" ice layers exceeding a 1 m thickness are encountered. The model is initialized in April 2009 with linearly thickening firn with elevation (0.14 m m$^{-1}$) on top of solid ice above the long-term equilibrium line altitude at ca. 1550 m ASL (Van de Wal et al., 2012). Subsurface calculations are performed on a 20 m

vertical grid with 0.25 m resolution (versus 0.5 m by Van As et al., 2012).

There is a lack of precipitation measurements over the Kangerlussuaq catchment of the ice sheet. Therefore our model includes a precipitation parameterization in which a constant precipitation rate is assumed for snowfall (air temperature below freezing) and rainfall (above freezing) when downward longwave radiation exceeds the blackbody emissions

calculated from air temperature. This rough estimate is partly justified by the small impact precipitation has on the outcome of this study because of the dry climate of the Kangerlussuaq catchment (Van den Broeke et al., 2008; Johansson et al., 2015). The precipitation rate is tuned to provide optimal results in terms of winter accumulation.

Importantly, the model does not use surface albedo measured at the weather stations, as it is spatially heterogeneous while

highly influential in model calculations. Instead we use Moderate Resolution Imaging Spectroradiometer (MODIS) Terra MOD10A1 albedo within the Kangerlussuaq catchment, averaged over the 100 m elevation bins utilized by the model, after removing spikes. We also calibrated MODIS albedo to five years of daily albedo measured at the weather stations of the Greenland Climate Network (GC-Net) and the Programme for Monitoring of the Greenland Ice Sheet (PROMICE). This is



justified because MODIS on average underestimates albedo for solar zenith angles below ca. 74° (mean bias of 0.043), given the best linear fit

$$\alpha_{weather\ station} = \alpha_{MODIS} + 0.114 \cdot \cos\theta_{noon} - 0.032 \qquad (3)$$

where $\alpha$ is albedo and $\theta$ is the solar zenith angle. While for Van As et al. (2012) the daily MODIS resolution was an argument for running the SMB model in daily time steps, we recognize the need to resolve the daily cycle in ice sheet runoff. Therefore the current model version runs at an hourly time interval with a fixed daily albedo.

The various above-mentioned changes to the Van As et al. (2012) model impact the melt and runoff calculations mostly by increasing. The SMB model is not in any way tuned to match river discharge.

To test the model's performance, we compared its calculations of ablation and accumulation with in-situ measurements. Over the course of seven melt seasons, the accumulated SMB at any time is modelled within 1.5 m ice equivalent of the measured values at the weather station sites in spite of differences in elevation (Fig. 5). The model overestimates winter snow accumulation at low elevation; Van den Broeke et al. (2008) suggest that accumulation does not get recorded by low-elevation weather stations because snow mostly collects in the depressions between the ~5-10 m diameter ice hummocks. At high elevation, winter accumulation and summer ablation (melt) are overestimated, which partially cancel each other out in terms of SMB (Fig. 5). Note that observations as well as model results place the elevation at which the climatological mass budget is in balance (SMB + refreezing = 0) around 1800 m ASL for 2009-2015, which is ca. 250 m higher than reported for the 1990-2011 period, and similar to elevations in the top-ranking years 1995, 1999, 2003, 2007 and 2010 (Van de Wal et al., 2012).

**2.4 Kangerlussuaq catchment delineation**

Lindbäck et al. (2015) delineated the "Kangerlussuaq catchment" by means of hydraulic potential analysis, which is a steady state proxy for routing of subglacial water (Shreve, 1972). The catchment stretches from the ice sheet margin to the ice divide with a total area of ca. 12,000 km², or 0.7% of the total ice sheet area. This delineation method is superior to methods entirely dependent on surface slope (e.g. Van As et al., 2012) as meltwater in this region does not run over the surface until it reaches the ice sheet margin due to the abundance of moulins and crevasses (Yang et al., 2016). There is evidence of meltwater reaching the glacier bed at over 130 km from the ice sheet margin where the surface is 1840 m ASL (Doyle et al., 2014).

We subdivide the catchment area into 100 m elevation bins to match the output of the SMB model, allowing multiplication with calculated ice sheet runoff (in m) to derive volumetric units. We do not take the ice-free catchment of Watson River into account (Weidick and Olesen, 1978). Tundra makes up a small fraction (5%) of the total catchment area (Mikkelsen et





al., 2016), contributing on average less than 0.1 km$^3$ yr$^{-1}$ (in the order of 1%) to discharge from precipitation, not counting losses from evaporation.

## 2.5 Meltwater runoff delays

The time series of proglacial river discharge and ice sheet surface runoff enable the calculation of routing delays per elevation bin, introduced by meltwater transiting the supra-, en-, sub- and proglacial environments. We achieve this by finding the highest correlation between river discharge and catchment-total runoff at time $t$ $(R(t) = \sum_{n=1}^{N} R_n(t))$ after introducing different time delays for every elevation bin: $R_{delay}(t) = \sum_{n=1}^{N} R_n(t - d_n)$, shifting the $R_n$ time series in

elevation bin $n$ by $d_n$ (full) hours before summing. In calculating correlations, we loop through all possible $d_n$ values. We limit the search by setting $d_n \leq d_{n+1}$, i.e. it cannot take shorter for meltwater to transit from higher elevation, and $d_{n+1} - d_n \leq$ *36*, i.e. the added time delay is at most 36 h compared to that of the neighbouring lower elevation bin. For 18 elevation bins (covering 50-1850 m ASL, up to the observed maximum elevation of the surface runoff line) and 37 delay hours to test per bin, this requires $37^{18}$ calculations of correlation, indicating the need for simplification to reduce computing time. Therefore

we test routing delays only every second or third elevation bin, interpolating delays linearly for intermediate bins. We apply 1-, 2- or 4-hourly time steps depending on the elevation of the bin.

Optimal discharge delays are defined as the mean of those for which correlation falls within 0.01 of maximum correlation. If more than 100 solutions fulfil this criterion, we calculate the average of the cases with the 100 highest correlations. The

standard deviation serves as a measure of spread in the results. We determine the optimal routing delays for every June, July, August and September of every year with overlapping river discharge and ice sheet runoff time series (2009-2015). We also determine the optimal delays for the entire seven-year time series. This correlation-based method is insensitive to errors in the magnitude of ice sheet runoff and river discharge (and thus in the stage-discharge relation), as correlation is a measure of covariability.

## 2.6 Hypsometric amplification

In deriving an equation to quantify hypsometric amplification of ice sheet discharge, we assume that river discharge can approximate the product of meltwater $M$ (in vertical units) and surface area $A$, integrated over the melt zone of the ice sheet

ranging from the margin at $Z_0$ up to the upper melt limit at $Z$. Further assuming that: 1) $M$ is a linear function of (positive) temperature $T$ and multiplier $F_M$, 2) $T$ is a linear function of elevation $z$ using linear lapse rate $\lambda$ and $T(Z_0) = T_0$, and 3) $A$ is a power function of surface elevation with multiplier $F_A$ and exponent $p$, we find:





$$D \approx \int_{Z_0}^{Z} M \cdot A \ dz \approx \int_{Z_0}^{Z} F_M \cdot T \cdot F_A \cdot (z - Z_0)^p \ dz \approx \int_{0}^{Z} F_M \cdot (T_0 + \lambda \cdot z) \cdot F_A \cdot z^p \ dz = F_M \cdot F_A \cdot$$

$$\left( \frac{T_0}{p+1} + \frac{\lambda \cdot Z}{p+2} \right) \cdot Z^{p+1} = \frac{F_M \cdot F_A}{(-\lambda)^{p+1} \cdot (p+1) \cdot (p+2)} \cdot T_0^{p+2} = F_D \cdot T_0^{p+2} \tag{4}$$

Thus, river discharge can be approximated from air temperature $T_0$ at $Z_0$ provided the ice sheet hypsometric amplifier $p$ is
known along with the melt ($F_M$) and area ($F_A$) factors. For the Kangerlussuaq region, air temperature is measured at the
approximate elevation of the lowest glacier margin (ca. 50 m ASL; Cappelen, 2016). We derive $p$ and $F_A$ from the area-
elevation distribution of the Kangerlussuaq catchment as delineated by Lindbäck et al. (2015). We also determine $p$ for the
entire Greenland ice sheet and its northern (> 77.13° N), southern (< 68.74° N), and central western (> 40.38° W) and
eastern (< 40.38° W) quarters, using the Bamber et al. (2013) surface digital elevation model. With nearly all variables in Eq.
(4) known, it can be solved for melt factor $F_M$, provided a representative free-atmospheric lapse rate is chosen.

## 3 Results

### 3.1 Surface runoff and river discharge variability

Ice sheet surface melt in the Kangerlussuaq ice sheet catchment, and thereby Watson River discharge, is confined to the May
to September period, with minor episodic exceptions (Figs. 6 and 7). River discharge in our observational period (2006-
2015) peaks between 11 July and 2 August each year (Table 1), with a median of 25 July. Peak discharge ranges by a factor
2.7 from a low peak of $1.2 \cdot 10^3$ m$^3$ s$^{-1}$ in 2015 to a high peak of $3.2 \cdot 10^3$ m$^3$ s$^{-1}$ in 2012. Annual totals range by a factor of 3.0
from 3.8 km$^3$ in 2015 to 11.2 km$^3$ in 2010 (Table 1, Fig. 6). These values match those of ice sheet runoff within uncertainty;
both time series show an equally large in inter-annual variability (Fig. 6b). Also intra-annual variability in river discharge is
large, as values can increase by over a factor of two over the course of a few days (Fig. 7). Discharge peaks roughly double
their normal values for the time of year are explained by intense ice sheet melt episodes, such as those in late summer 2011
(Doyle et al., 2015) and mid-July 2012 (Fausto et al., 2016; Mikkelsen et al., 2016). Diurnal variability in river discharge is
on the order of 200 m$^3$ s$^{-1}$, or 10-20% of the total signal. Meanwhile, Kangerlussuaq catchment ice sheet melt regularly
exceeds 3000 m$^3$ s$^{-1}$ during mid-day while often halting at night, thus displaying a 10-20 times larger diurnal variability than
river discharge.

Many other aspects of the runoff and discharge time series can be identified in Fig. 7, such as the influence of rain or rapid
drainage by supraglacial and ice-dammed lakes. We will return to these topics in the discussion section. For now, we focus
on the science questions that arise from the results in terms of spatial and temporal variability:

1) Can we explain the large, factor three inter-annual variability in annual total discharge in terms of hypsometric
   amplification, and can we quantify this effect?





2) Can we quantify the factor 10-20 moderation of meltwater release by the ice sheet and interpret results in terms of routing delays in the supra-, en-, sub- and proglacial environments?

## 3.2 Hypsometric amplification

To investigate the hypsometic amplification through increases in ice sheet area with elevation, we return to Eq. (4) that allows us to quantify the temperature response of ice sheet discharge provided we determine the value for hypsometric amplifier $p$. For a linearly surface slope $p = 0$, while $p < 0$ for a convex hypsometry typical for mountain glaciers. The higher the $p$ value, the more sensitive the ice mass is to atmospheric temperature increase. Hypsometric amplification ($p > 0$) is

known to occur for the Greenland ice sheet (Van As et al., 2012; McGrath et al., 2013; Mikkelsen et al., 2016).

From the geometry of the Kangerlussuaq catchment (Lindbäck et al., 2015) we deduce that $A = F_A \cdot (z - Z_0)^p \approx 160 \cdot (z - 50)^{1.4}$ is a good ($r = 0.993$) approximation for the area below 1350 ASL, that generated 85% of all surface runoff for 2009-2015 according to our model calculations. Correlation reduces to $r = 0.80$ when including the nearly twice as narrow

region between 1350 m and the maximum observed runoff elevation around 1850 m (Fig. 1). A $Z_0$ sensitivity test indicates that $p$ falls in the range 1.2-1.6 for the Kangerlussuaq catchment. An independent method using the elevations of six on-ice weather stations and their distance to the margin gives $p = 1.3 \pm 0.2$, ignoring changes in catchment width with elevation. Plotting Fig. 4 with logarithmic axes confirms the relation between river discharge and air temperature with exponent $p = 1.4$.

For the entire Greenland ice sheet we find that $p = 1.5$ (Fig. 8) for elevations up to 2550 m ASL, well above the runoff area. This $p$ value indicates that the Kangerlussuaq catchment can be considered a representative section of the ice sheet in terms of its temperature sensitivity of meltwater release. At higher elevations the ice sheet converges towards its topographic peaks, altering the area-elevation distribution, yet not of relevance until runoff starts occurring much higher on the inland

ice. Having divided the ice sheet in four equal portions, we find that $p = 1.5$ in the north ($> 77.13°$ N), $p = 1.3$ in the south ($< 68.74°$ N, including the Kangerlussuaq catchment), and $p = 1.8$ for the western slope of the ice sheet (Fig. 8). The largest hypsometric amplification is found for the eastern slope of the Greenland ice sheet ($p = 2.4$). This relatively high factor is likely caused by the eastern ice sheet receiving higher accumulation rates, and by it being bordered by high mountains, forcing the ice sheet to converge into valley glaciers and thus a smaller area at lower elevations, as opposed to the generally

less irregular margin elsewhere in Greenland. Nevertheless, the hypsometric amplifier of 2.4 does suggest that increases in temperatures in east Greenland yield the largest increases in ice sheet meltwater discharge into the oceans.

To determine the impact of the hypsometric amplifier, we calculate using Eq. (4) how much temperature variability is required to produce a factor 3.0 discharge variability for $p = 1.4$. Applying this temperature variability to a reference scenario





with $p = 0$, we find that the hypsometry of the Kangerlussuaq catchment amplifies meltwater release by 56%. Extrapolating this methodology to other regions of the ice sheet, we find a hypsometric amplification of 62% for the entire ice sheet ($p = 1.5$), and a 115% amplification for the eastern slope of the Greenland ice sheet ($p = 2.4$).

Figure 4 features the estimated temperature response of Watson River discharge using the $p$ value of 1.4, and a discharge factor $F_D$ for bare ice (July-September) of 0.31. Applying a mid-range free-atmospheric adiabatic lapse rate $\lambda$ of $-7 \cdot 10^{-3}$ °C m$^{-3}$, we can solve Eq. (4) to find melt factor $F_M = 8.2$ mm water equivalent °C$^{-1}$ day$^{-1}$, which can be considered the catchment-average positive degree day factor. This factor is only of relevance to the gap filling of the Watson River discharge time series; it does not influence this study's main conclusions. Note that melt is estimated from atmospheric temperature outside the surface-controlled, shallow stable boundary layer over the ice sheet, which is argued to be preferred as it better represents atmospheric forcing of melt (Lang, 1968; Ohmura, 2001).

Although the Watson River discharge time series confirms the value of the hypsometric amplifier in the Kangerlussuaq catchment, the uncertainty in determining river discharge from air temperature (i.e. not the topic of this study) remains large at a conservative 70% (Fig. 4). Part of the scatter in Fig. 4 is due to the fact that meltwater generated at the ice sheet surface takes time to transit the supra-, en-, sub- and proglacial routing environments, which is not properly accounted for in the figure. Therefore we turn towards our second science question, on quantifying such routing delays.

### 3.3 Routing delays

Watson River discharge trails ice sheet surface meltwater runoff by several days (Figs. 6a and 7) caused by routing delays in transiting the supra-, en-, sub- and proglacial environments. Figure 9 illustrates in grey the average routing delay per elevation bin for which the ice sheet surface meltwater best matches river discharge in terms of correlation ($r = 0.90$-$0.91$). Routing delays increase with elevation, and more so at higher elevation where the distance to the Watson River bridge measurement site becomes increasingly large due to the widening of the elevation bins. The dependency of the delay ($t_d$ in h) on surface elevation ($z$ in m ASL) can be approximated ($r = 0.997$) by the polynomial

$$t_d = 65.6 \cdot 10^{-6} \cdot z^2 - 18.1 \cdot 10^{-3} \cdot z + 7.3. \tag{5}$$

This delay also includes the transit through the 26-33 km long proglacial river system, which we estimate to typically take 4-5 h based on measurements taken during a 2010 ice-dammed lake drainage (Mikkelsen et al., 2013) and is thus small compared to the total delay, which exceeds one week at 1800 m ASL. We calculate the average effective, horizontal travel velocity through the supra-, en- and subglacial environments to be $0.7 - 0.8$ km h$^{-1}$ for water originating from the majority (>90%) of the runoff area (650-1850 m ASL). This spatially rather constant water velocity, indicates that for the multi-year average time delay as presented by Eq. (5), hydraulic efficiency is similar throughout most of the runoff area of the ice sheet.



Our velocity values are below the average but within the $0.2 - 1.6$ km h$^{-1}$ range reported by Chandler et al. (2013) and Cowton et al. (2013), who used a methodology that bypasses supraglacial routing. Mikkelsen et al. (2016) found a general discharge delay of between one and five days, which we confirm for the ice sheet area between 650 and 1450 m ASL, making up a dominant portion of 58% of the runoff area.

The routing delay Eq. (5) is used in Fig. 7 to adjust the hourly catchment-total values in light blue to delayed runoff in red. We apply an additional 10-h smoothing to adjust the meltwater runoff record from being a composite of discrete elevation bins to something that also represents the spread in routing delays within the 100 m elevation bins. Delayed runoff matches river discharge in terms of amplitude and variability, especially during the low discharge years 2009, 2013 and 2015. However, during certain periods agreement is lower. For instance, in 2012 the delay in ice sheet runoff is too small before the peak of the melt season and too large during and after, likely related to a rapid development of the englacial drainage system (Bartholomew et al., 2011; Palmer et al., 2011). To discover intra-seasonal changes in routing delays and thus hydraulic efficiency of the ice sheet, we also apply the correlation procedure to each June-September month in the 2009-2015 period.

Figure 9 illustrates that the elevation-dependent delays display a large range, with delays during some years being 2-3 times larger than during other years. June routing delays are, almost exclusively, larger than the multi-year average (Fig. 9a), because of slow percolation through winter-accumulated snow on the ice sheet surface (Bøggild et al., 2005), and an underdeveloped englacial drainage system. In July, commonly the peak river discharge month (Table 1), routing delays and the spread therein are smaller as surface snow is largely melted and the englacial drainage system develops rapidly in response to increases in water supply. The reduced delays are most relevant at the lower and mid elevation bands from which most meltwater originates. Development of the englacial drainage system can occur over the course of mere days; for instance in the first half of July 2012, the drainage system shifted from below-average efficiency (larger delays in Fig. 9a) to above-average (smaller delays in Fig. 9b). After the peak of the melt season, in August, the englacial drainage system remains capable of efficiently routing the dropping water volumes given the fact that delays are typically similar to those in July (Fig. 9c). In September, routing delays increase as drainage channels close and hydraulic efficiency reduces, most notably at lower and mid elevation where hydraulic efficiency is rapidly lost as water supply diminishes (Fig. 9d).

We note that not all optimal solutions at monthly time intervals represent glacial drainage delays accurately, because both the ice sheet runoff and river discharge time series include features of other processes. For instance, the August and September 2010 delays are unrealistic due to a modelled melt underestimate during extreme melt periods (Fausto et al., 2016), and an ice-dammed lake drainage that cannot be captured by the model. Likewise, August and September 2014 delays are inaccurate due to overestimated ice sheet runoff during a rain event (see Discussion section). We therefore regard the monthly panels in Fig. 7 as ensemble solutions, and do not overemphasize results from specific months. However, some



outliers are realistic, such as the July 2012 delay at elevations above 1200 m ASL, when hydraulic efficiency was higher than in any other year, as a consequence of unprecedented melting even at the highest plotted elevation bands (Mikkelsen et al., 2016).

## 4 Discussion

### 4.1 Record-setting discharge July 2012

The highest discharge measured at Watson River occurred during the period 10-14 July 2012. Presumably this was the highest discharge in nearly 60 years given the 1955 Watson bridge road dam washout on 11 July 2012. This high discharge event coincided with a large melt episode impacting the entire Greenland ice sheet (Nghiem et al., 2012). With the new rating curve, we determine that relatively, the July 2012 discharge peak is reduced from roughly quadruple to double its normal mid-range value (Fig. 7). In determining the causes of the extreme discharge, in addition to high ice sheet melting, we look into the timing of event. The KAN weather stations (Fig. 1) reveal that regionally the high melt episode started around 11:00 UTC on 8 July and lasted 3.5 days. Watson River reached its peak stage 3.3 days after the start of high melt. Calculating optimal discharge delays for the period 9-15 July, encompassing the extreme discharge event, we find that in 3.3 days the ice sheet melt water could have travelled from as high as 1500 m ASL (Fig. 9b). In other words, all meltwater passing at the bridge during the bridge road dam washout originated from the long-term ablation area (defined in Van de Wal et al., 2012). Only towards the end of the 10-14 July extreme river discharge did ice sheet contributions from the upper runoff area (ca. 1800 m ASL) arrive at the bridge site. Therefore we consider the role of largely impermeable ice layers in the firn (Machguth et al., 2016) in generating peak river discharge and road dam washout (Mikkelsen et al., 2016) to be minor.

In our interpretation, it appears relevant for the peak-discharge event that uninterrupted high melt persisted for a period of more than three days (Fausto et al., 2016), allowing the (hypsometrically amplified) meltwater totals generated in the upper ablation and lower accumulation area on 8 and 9 July to exit the ice sheet along with that generated closer to the margin 2-3 days later (Fig 4b). Figure 7g illustrates that transit times for the upper elevations were lower than in other years. We speculate that this is partly because the 2012 peak melt was preceded by another high-melt episode around day of year 175 (dark blue line in Fig. 7g), increasing the efficiency of the englacial drainage system, allowing for faster transit during the following high-melt episode.

Altogether, we find the keys to record-setting discharge in Watson River to be 1) intense ice sheet melting that 2) continues for several days, 3) amplified by ice sheet hypsometry and 4) is preceded by another high-melt episode. Because of the



multi-day transit time for meltwater to reach the community of Kangerlussuaq, an early warning system of future bridge floodings could in principle be deployed, relying on in-situ ice sheet melt measurements.

## 4.2 Stage-discharge relation and road dam washout

The washout of the road dam at the Watson River bridge during the July 2012 peak discharge event potentially presents a challenge for deriving a single stage-discharge relation that applies to water levels recorded before the event, and after, when the newly formed third channel became a bridge segment (Fig. 2). For water levels below ca. 7 m over our lowest pressure transducer (Fig. 3), channel 3 remains dry and no change in the rating curve is expected, but for higher river stages and thus discharge values exceeding ca. 1200 m$^3$ s$^{-1}$, a change may have occurred. We have no indication that water stage at our pressure transducer site 150 m upstream of the bridge is affected by the road dam washout, yet it cannot be ruled out, mostly at extreme discharge levels that could have yielded upstream pooling in years before the washout. If a change in the rating curve due to the road dam washout occurred, we would expect overestimated river discharge at high river stage pre-2012. Judging from the 2010 and 2011 comparison of discharge with ice sheet surface runoff in Fig. 7, overestimated discharge at high stage is at least plausible. An argument for keeping a single rating curve for the entire 2006-2015 period is that although the curve is established using ADCP data retrieved after the road dam washout, the calculated discharge during pre-washout ice-dammed lake drainages agrees well with values reported by Russell et al. (2011). It is likely that the 15% uncertainty in our discharge calculation encompasses most or all of the changes inflicted by road dam washout and bridge segment construction. In any case, changing the rating curve and thus the amplitude of the discharge signal does not impact the primary conclusions of this study.

## 4.3 Rapid lake drainages

Although the total supraglacial lake volume that is released in drainage events is in the order of a few percent of the annual discharge for the Kangerlussuaq region of the ice sheet, rapid lake water release can contribute tens of percents to instantaneous discharge, primarily in the early melt season (Fitzpatrick et al., 2014; Mikkelsen et al., 2016). For instance, a surge in river discharge resulting from clustered supraglacial lake drainage was suggested to have occurred during days of year 180-184 in 2010 (Doyle et al., 2013; Hasholt et al., 2013). In this period, we find river discharge to exceed modelled ice sheet runoff as the latter does not take into account rapid lake drainage (Fig. 7e). From our records, we find no further evidence of a clustered drainage event of similar magnitude in the period 2009-2015, although smaller river discharge spikes around day of year 180 in other years (e.g. 2008, 2009, 2011) may very well also be related to supraglacial lake drainage (Bartholomew et al., 2011). Mikkelsen et al. (2016) determined the contribution of supraglacial lake drainage to annual Watson River discharge to be negligible.





Rapid drainages of the ice-dammed lake at the margin of the Russel Glacier, the northernmost glacier within the Kangerlussuaq catchment, produce discharge peaks lasting under two days (Russell et al., 2011). At least five such jökulhlaups can be identified towards the end of the melt season in 2007, 2008, 2010, 2012 and 2013 (Fig. 7). They contribute less than 1% to the annual-total discharge, yet they can yield high peak values. For instance, the 2007 jökulhlaup
resulted in a higher peak discharge than all melt-induced values in 2006-2015, except during high melt years 2010 and 2012.

## 4.4 Rain events

Figure 7 illustrates a good agreement between ice sheet surface runoff and river discharge, with a noteworthy exception for
the mid-July 2012 extreme discharge event when river discharge is seen to exceed runoff. Underestimating modelled ice sheet melting has been identified to be a potential issue during this unprecedented episode (Fausto et al., 2016), and could therefore explain mid-July mismatch in Fig. 7g. On other occasions, ice sheet runoff exhibits distinct peaks not entirely captured by the river discharge measurements in Fig. 7, such as in July 2010 (days 205-208) September 2013 (day 248) and 2015 (day 247), or most notably in late August 2014 (day 231-235). These peaks coincide with model-generated rainfall
across the entire elevation range. Rain events can yield intense ice sheet surface melting largely due to increases in longwave radiative and turbulent heat fluxes (Doyle et al., 2015; Van Tricht et al., 2016). Precipitation measurements in Kangerlussuaq (Cappelen, 2016) and near the ice sheet margin (Johansson et al., 2015) confirm rainfall in these periods, and positive temperatures yield the possibility of liquid precipitation over the ice sheet. Given that little snow is left on the ice sheet surface to retain water towards the end of the melt season, the modelled rain and enhanced meltwater produce distinct runoff
peaks (dark blue lines in Fig. 7). Delayed runoff values (red lines) exceeding river discharge following these rain events indicate that rainfall is likely overestimated during these often short-lived events, most notably in late August 2014. Other rain events such as in late August 2011 (Doyle et al., 2015) are modelled accurately (Fig. 7f). Climatologically, Kangerlussuaq is dry in terms of precipitation due to blocking topography to the southwest (Van den Broeke et al., 2008; Johansson et al., 2015).

## 4.5 Piracy between catchments

Disparity between river discharge and ice sheet runoff may also be related to transient behaviour between adjacent catchments, driven by seasonal changes in basal water pressure. Lindbäck et al. (2015) found that at ice sheet elevations
above ca. 1200 m ASL changes in the subglacial hydrology can lead to large shifts in the Kangerlussuaq ice sheet catchment boundaries during the melt season, a.k.a. piracy between catchments. Since piracy impacts the meltwater running off from the upper ablation area and above, the effect of a catchment boundary shift on catchment-wide runoff is expected to be largest during high-melt periods when a substantial amount of meltwater is generated at high elevations. However, Lindbäck et al. (2015) find that the Kangerlussuaq catchment above 1200 m ASL shift norths in its entirety when subglacial water




pressure builds, causing only a small change in surface area and therefore little (ca. 10%) increase in catchment-wide runoff. Yet it remains plausible that the mismatch between river discharge and ice sheet runoff during the high-melt seasons of 2010-2012, and particularly during the mid-July 2012 extreme melt episode, is in part explained by a temporarily underestimated catchment area at high elevations.

**4.6 Retention in firn and englacial environment**

Figure 7 illustrates that modelled ice sheet runoff exceeds the river discharge values by c. 1000 $m^3$ $s^{-1}$ in spring of most years. We attribute this to a model underestimate of meltwater retention in winter-accumulated snow and an increase in

meltwater storage in supraglacial lakes (Fitzpatrick et al., 2014) that is not captured by the model. Both processes provide plausible explanations given the better agreement between river discharge and ice sheet runoff in summer and autumn when winter snow has melted and most supraglacial lakes have drained. Since the mismatch is smallest for spring 2010 when winter accumulation was below average (Tedesco et al., 2011), it is most likely that the SMB model underestimates retention in snow, possibly due to underestimated snow accumulation that does not get recorded by weather stations when it collects in

crevasses and in between ice hummocks in the lower ablation area (Van den Broeke et al., 2008).

In all, river discharge and ice sheet runoff agree at an unprecedented level in this study because of 1) using an improved, validated, observation-based time series of modelled ice sheet runoff, 2) studying a large catchment area implying a reduced relative uncertainty in delineation (Lindbäck et al., 2015), 3) constraining the river discharge calculations with superior

ADCP measurements, and 4) the introduction of meltwater routing delays. In light of the high level of agreement in terms of variability and quantity, and the fact that we are able to provide plausible explanations for periods of mismatch, we find no evidence of meltwater storage in the en- and subglacial environments in amounts that surpass the detection limit as set by our methodological uncertainties. Using similar methods, such retention has been suggested to be significant for the Greenland ice sheet in previous studies (Rennermalm et al., 2013; Overeem et al., 2015; Smith et al., 2015; Mikkelsen et al., 2016) with

values of up to half the meltwater availability reported. Whereas changes in (supra- and) subglacial storage are known to occur in Greenland as seen from rapid lake drainages, they are reported to make up for only a few percent of the annual discharge in the Kangerlussuaq region (Fitzpatrick et al., 2014; Palmer et al., 2015), i.e. well below our uncertainty and thus detection level for ice sheet runoff and river discharge.

**5 Conclusions**

Watson River in west Greenland drains a ca. 12,000 $km^2$ sector of the ice sheet, where the altitude at which the climatological mass budget is in balance (SMB + refreezing = 0) has increased from ca. 1550 m ASL for 1990-2011 to ca. 1800 m ASL for 2009-2015. We calculate ice sheet runoff and river discharge for a seven- and ten-year period, respectively,



using an improved, validated, observation-based ice sheet SMB model and an updated river stage-discharge relation constrained by newly available ADCP measurements.

Interannual variability in ice sheet meltwater release is found to be large; for instance river discharge ranges from 3.8 km$^3$ in 2015 to 11.2 km$^3$ in 2010, a factor 3.0 difference. With hypsometry known to be an amplifier of ice sheet runoff, we deduce that discharge $D$ can be approximated using regional air temperature $T_0$ using $D \sim T_0^{p+2}$. Here $p$ is the hyprometric amplifier, determined to be 1.4±0.2 for the Kangerlussuaq catchment of the ice sheet. For $p = 1.4$ we calculate ice sheet meltwater release to be amplified by ca. 56% due to hypsometry. We determine $p = 1.5$ for the entire Greenland ice sheet, with regionally higher values and thus higher climate sensitivity, such as a value of $p = 2.4$ for the eastern slope of the ice sheet.

Diurnal variability in river discharge (ca. 100 m$^3$ s$^{-1}$) is found to be more than an order of magnitude smaller than the variability in ice sheet surface meltwater runoff for the Kangerlussuaq catchment. The difference in diurnal variability is a result of the time lag involved in routing meltwater from its origin at the ice sheet surface, through the supra-, en-, sub-, and proglacial environments to reach the river monitoring site. Introducing time lags to ice sheet runoff as a function of elevation results in a good agreement with river discharge. Optimal delays reveal considerable changes in ice sheet hydraulic efficiency throughout the melt season, with time lags smallest shortly after high melt episodes that overwhelm and develop the englacial drainage conduits. On average, the routing delays can be approximated by $t_d = 65.6 \cdot 10^{-6} \cdot z^2 - 18.1 \cdot 10^{-3} \cdot z + 7.3$, which for instance yields that meltwater generated at 1500 m ASL takes 5-6 days to be released from the ice sheet. An implication of this result is that melt episodes lasting several days are more likely to cause overbank river flooding, such as at Kangerlussuaq in mid-July 2012. Finally, due to the close agreement between river discharge and ice sheet surface meltwater runoff after the inclusion of routing delays, we find no evidence of meltwater retention in the en- and subglacial environments beyond the detection limit set by our methodological uncertainties.

**Author contribution.** D. van As and B. Hasholt conceived the study; B. Hasholt, A. B. Mikkelsen and D. van As monitored river discharge; M. H. Nielsen provided ADCP data; K. Lindbäck delineated the catchment; L. C. Liljedahl enabled the in-situ monitoring of the ice sheet; D. van As calculated river discharge, ice sheet runoff and wrote the text with contributions by all coauthors.

**Acknowledgements** Over the years the Watson River discharge monitoring has been (co)financed through various funding sources: the Commission for Scientific Research in Greenland grants 07-015998, 09-064628 and 2138-08-0003, the Danish Natural Science Research Council grant 272-07-0645, the Center for Permafrost (CENPERM.ku.dk), the Department of Geosciences and Natural Resource Management (IGN.ku.dk), the Greenland Analogue Project (GAP) and the Danish Energy Agency (www.ENS.dk). The weather stations monitoring the ice sheet and SMB modelling are financed by the GAP with contributions from the Programme for Monitoring of the Greenland Ice Sheet (www.PROMICE.dk). Kangerlussuaq



meteorological data are collected by the Danish Meteorological Institute (www.DMI.dk). We acknowledge local support by the Centre for Ice and Climate (www.iceandclimate.nbi.ku.dk), CH2M HILL Polar Services (www.CPSpolar.com), Kangerlussuaq International Science Support, and the Greenland Survey (www.Asiaq.gl). Support various in nature was kindly provided by Andreas Ahlstrøm, Sam Doyle, Emily Henkemans, Alun Hubbard (GAP subproject A lead), Anne Kontula (GAP co-lead), Horst Machguth, Sebastian Mernild, Paul Smeets, Larry Smith, Kisser Thorsøe and many others.

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





Table 1. Watson river discharge values with uncertainty. Values marked by * originate from an ice-dammed lake drainage event.

| | 2006 | 2007 | 2008 | 2009 | 2010 | 2011 | 2012 | 2013 | 2014 | 2015 |
|---|---|---|---|---|---|---|---|---|---|---|
| Total discharge ($km^3$) | 5.4 ±2.9 | 7.5 ±1.2 | 5.5 ±0.9 | 4.9 ±0.9 | 11.2 ±1.7 | 7.8 ±1.2 | 10.7 ±1.6 | 4.3 ±0.7 | 6.8 ±1.0 | 3.8 ±0.6 |
| Peak discharge ($10^3$ $m^3$ $s^{-1}$) | 1.69 ±1.19 | 2.21* ±0.33 (1.99 ±0.30) | 1.29 ±0.19 | 1.43 ±0.21 | 2.38 ±0.36 | 1.94 ±0.29 | 3.22 ±0.48 | 1.44 ±0.22 | 1.59 ±0.24 | 1.18 ±0.18 |
| Date of peak discharge (UTC) | 26 Jul | 31 Aug* (13 Jul) | 1 Aug | 18 Jul | 31 Jul | 24 Jul | 11 Jul | 2 Aug | 26 Jul | 24 Jul |

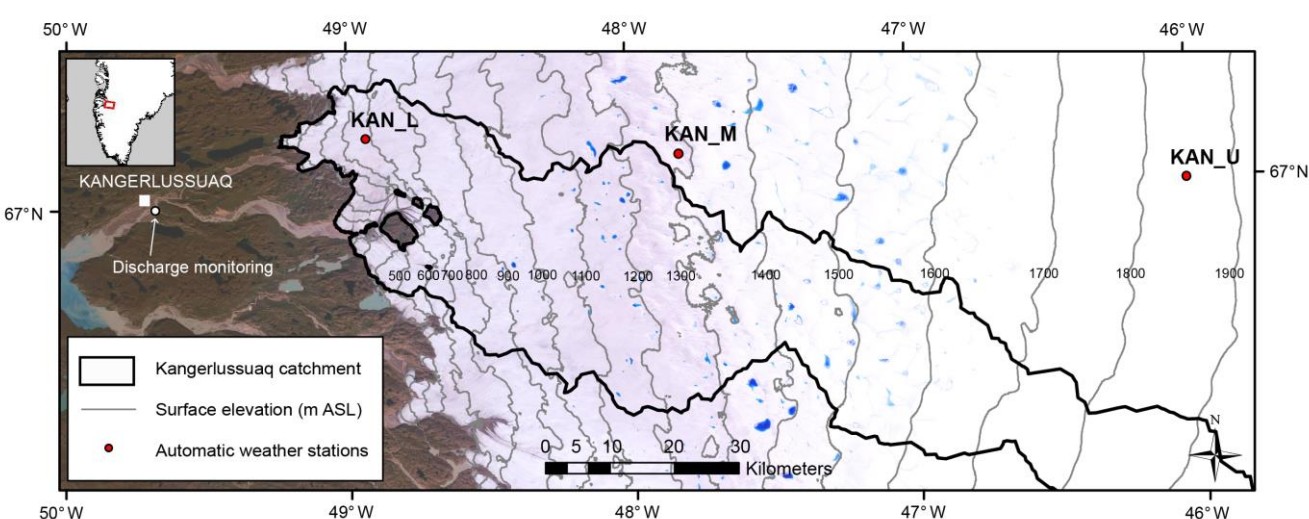

Figure 1: Map with the location of Kangerlussuaq, the monitoring site of Watson River discharge. Also plotted are the locations of the on-ice weather stations and the catchment delineation used in the calculation of ice sheet surface meltwater discharge. The inset illustrates the location of the Kangerlussuaq region in Greenland.




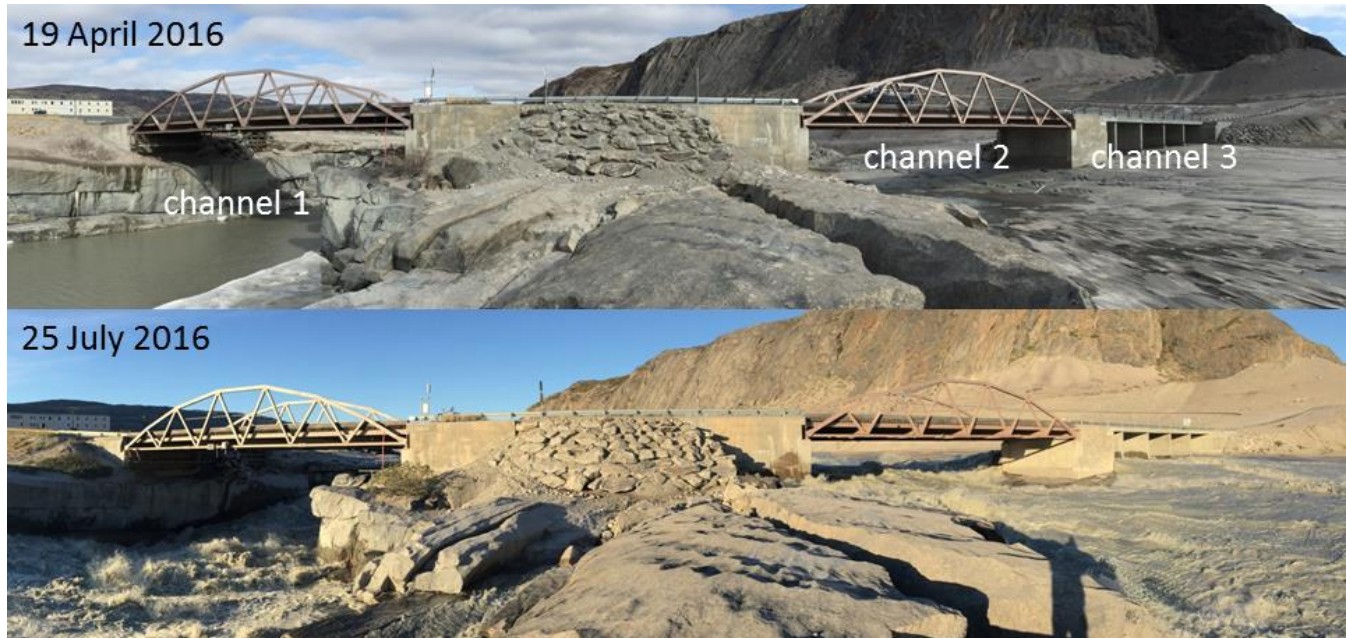

Figure 2: Pictures of the three channels underneath Watson River bridge at low (top) and high discharge (bottom).

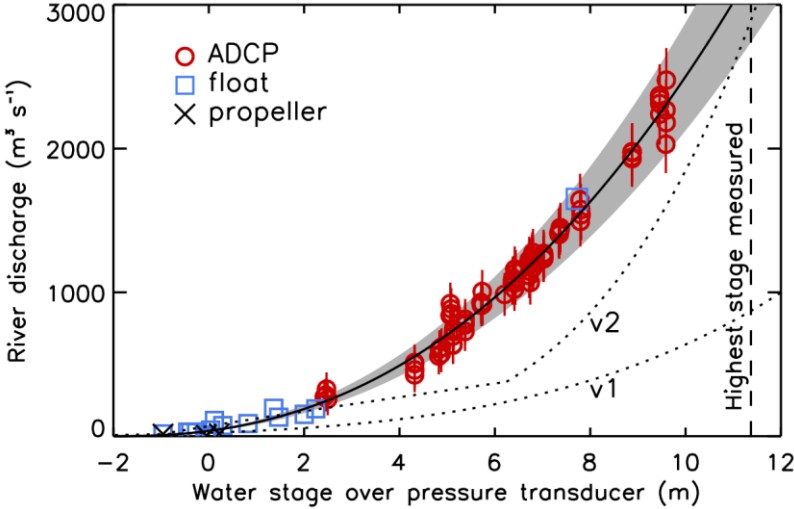

5  Figure 3: Measured Watson River discharge versus stage. Symbols denote discharge measurements by three methods. The

black line represents the best power-function fit, and the grey area a 15% uncertainty range. Dotted lines illustrate earlier

versions (v1 and v2) of the stage-discharge relation.




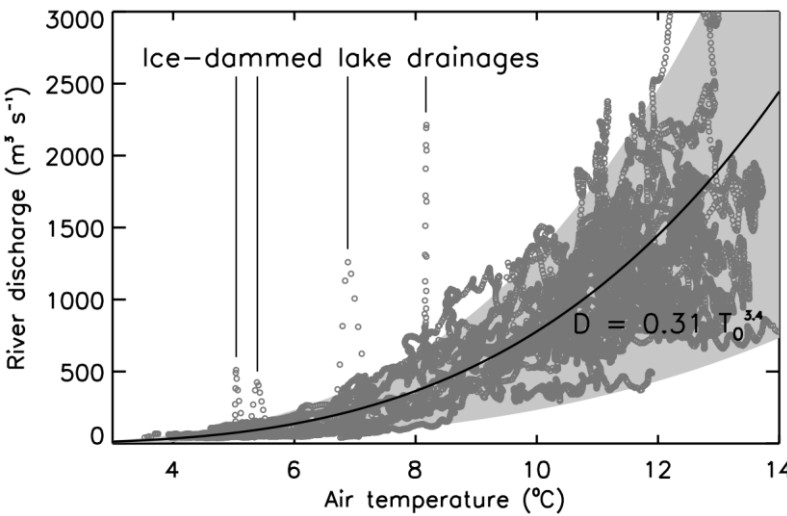

Figure 4: Hourly values of river discharge versus air temperature in Kangerlussuaq for July-September data 2008-2015 (dark grey). The solid line denotes a power-function fit to the data, with the light grey area the 70% uncertainty range. Ice-dammed lake drainages ("jökulhlaups") stand out as spikes.

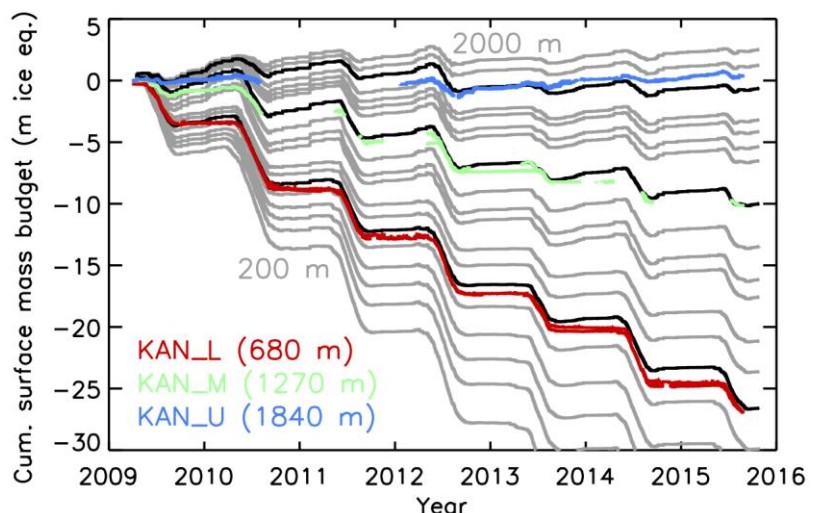

Figure 5: Modelled (grey) and measured (colours) cumulative surface mass balance in the Kangerlussuaq catchment of the Greenland ice sheet. Model values are given for every 100 m elevation bin (200-2000 m ASL). Black lines denote the bins closest to the "KAN" weather station elevations, i.e. not at the exact same elevation. Observation time series are composites 10 of two sensors; note that winter accumulation at KAN_M is not captured by one of the two.





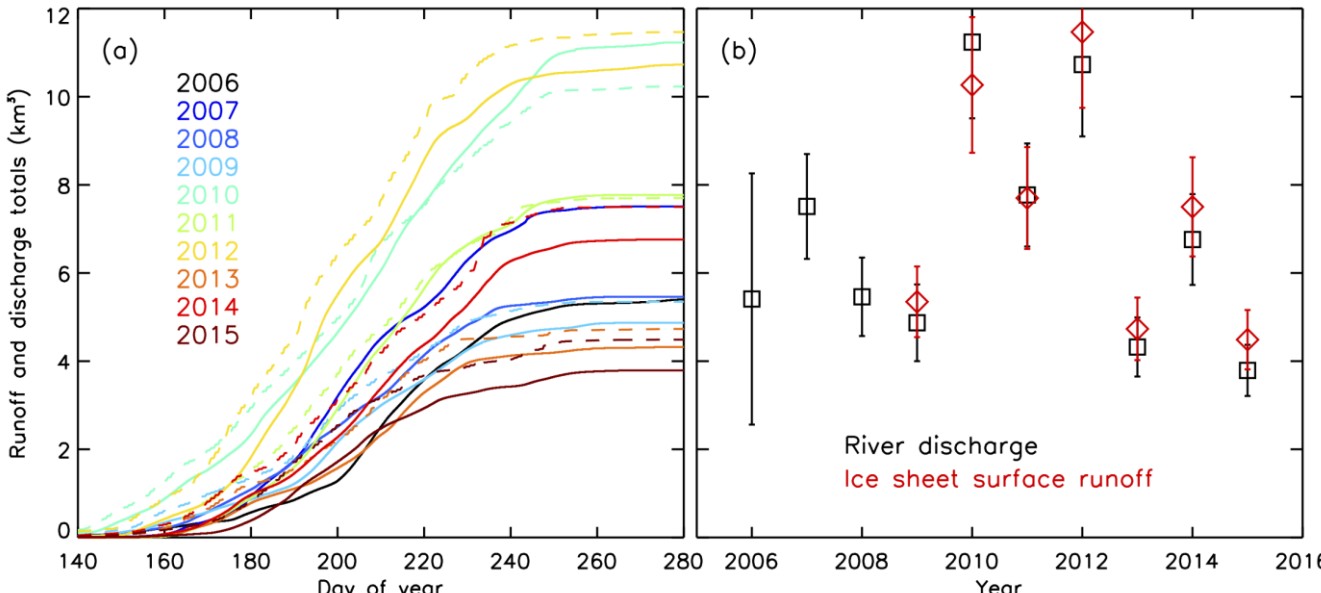

Figure 6: (a) Cumulative Watson River discharge (solid lines) and ice sheet surface meltwater runoff from the Kangerlussuaq catchment (dashes lines). (b) Annual totals and uncertainty of discharge (black) and runoff (red).





Figure 7: Hourly Watson River discharge (black), and hourly (light blue) and daily (dark blue) catchment-total ice sheet surface meltwater runoff. The red line gives meltwater runoff with elevation-dependent routing delay. Dark grey lines are estimated from temperature (see Methods section). Light grey areas represent discharge uncertainty.





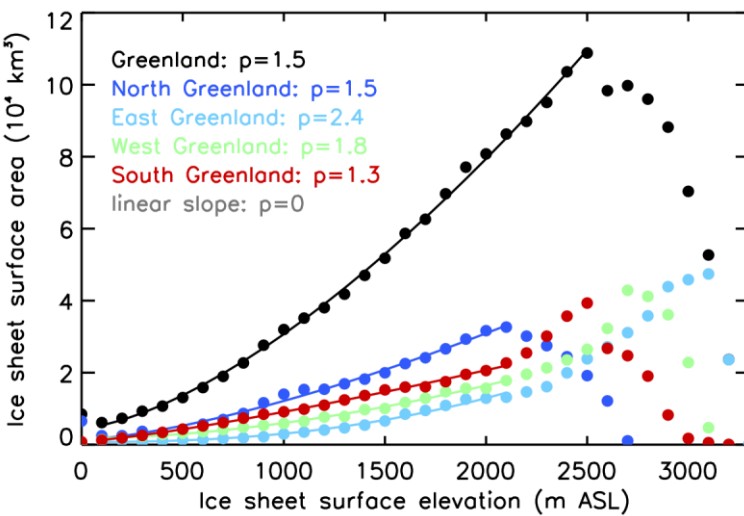

Figure 8: Area per 100 m elevation bin of the Greenland ice sheet for the entire ice sheet (black dots), and subsections (colored dots). Lines represent the least-squares power-law fits.







Figure 9: Optimal meltwater routing delays for the supra-, en-, sub- and proglacial environments determined for June (a), July (b), August (c) and September (d). As a reference, grey illustrates the values for the entire period with overlapping time series of river discharge and ice sheet surface meltwater runoff (June 2009 – September 2015), as used for the red line in Fig. 7. The dashed line illustrates optimal routing delays for the 9-15 July 2012 period, encompassing the 10-14 July extreme discharge event.