# Peer review of "Hypsometric amplification and routing moderation of Greenland ice sheet meltwater release"

_The Cryosphere, 2016_

## Referee Comment (RC1) · X. Fettweis (Referee) · 26 Jan 2017

This paper presents a very interesting update of the van As et al. (2014) paper discussing river discharge over GrIS and successfully explains here its temporal variability with the help of modelling. This paper is well written, discusses a lot of interesting stuff (routing delays, melt retention, rapid lake drainage, rainfall events, . . .), deserves to be accepted in TC as a robust following of the van As et al. (2014) paper and can be published with only some minor revisions for me.

- the paper should more highlight that the considered catchment (in a very dry area) is likely not representative of other GrIS areas (for meltwater retention, lake drainage, . . .).

- An interesting sensitivity experiment to evaluate the retention in firm (Section 4.6)

should be to increase the winter snowfall by a factor 2. While the agreement is very good with obs, higher winter accumulation and higher melt could give similar results. Therefore, it is for me a bit too early to claim that there is not meltwater retention in this catchment. This should be confirmed by sensitivity experiments (e.g. Snowfall x 2 + Melt x 1.5).

- Daily MAR outputs could be used in addition to check that melting routing delay used here are not too model dependant because it is very likely that the melting routing delays used here (Fig9) could compensate biases in the model. MAR has also its own but different biases. Evaluating how the routing delays is model dependant will add robustness in the paper. I can provide 7.5km daily outputs to the authors if they find that it is an interesting addition to their paper.

- What does "ca." means ? It is used several times in the paper.

---

## Referee Comment (RC2) · Anonymous Referee #2 · 15 Mar 2017

General comments

This study examines how routing of melttwater from the surface of the ice sheet delays meltwater on its way to the river outlet at the fjord. A unique time series of river discharge data measured at the Watson River is used to quantify the meltwater output. To quantify ice sheet runoff, a three automated weather stations along an elevation gradient are use to force a surface mass balance model. The authors quantify that meltwater routing can delay outflow with up to a week at the highest elevations.They find a good match between ice sheet runoff and proglacial river discharge suggesting that meltwater retention is insignificant at larger scales.

This paper is an important contribution to the literature about Greenland ice sheet runoff and meltwater losses. I urge the authors to provide a more convincing analysis

of about the impact of rainfall on modeled runoff (see specific point 8). Other than that, most comments are suggestions to clarify the paper for readers. The work is very interesting and the paper will get a wide readership.

Specific comments

1. Regarding the rating curves (section 2.1). A more in-depth discussion about the previous ratings curves are needed. Were they created with some of the 90% of the float discharge measurements that now are discarded? Explain why they are so different. 2. Clarify what uncertainties are considered in the uncertainty estimates made with each the three discharge methods. 3. Regarding the gap filling method (section 2.2). Consider referring to the positive degree-day melt model as a motivation for using temperature for gap filling. Also, please provide the correlation/coefficient of determination and/or RMSE for fit between the observations and the temperature-based model. 4. Alternatively, forego the temperature based gapfilling method all together. The paper would be simplified if the SMB model output with runoff delays would be used for gap filling the river discharge time series rather than the temperature based model. The advantage of the SMB model is that it physically based. This hinges upon that the runoff delay can be developed without the gap-filled time series 5. Regarding testing the SMB model performance (section 2.3). It would be good to quantify the difference between model estimates and the in situ ablation and accumulating data. 6. Specify the uncertainties in the catchment delineation method and discuss the implications on the results and conclusions. For example, consider work by Ben Hudson about how DEM uncertainty propagate to catchment delineation (see ref by Carroll et al.) 7. It is unclear how science question 1 is examined in section 3.2. The section appears to quantify the effect of the factor three inter-annual variability on hypsometric amplification, but does not appear explain it. It is a difficult section to follow so it may be in there, but difficult to follow. Consider rewriting. 8. The analysis about rainfall events and their impact on ice sheet meltwater production presented in the discussion warrants some deeper analysis. First, provide a more comprehensive identification of these events

and how often they coincide with rainfall events. Second, to prove your argument that the temporal mismatch between the red and black lines in Figure 7 can be explained by rainfall events, run the SMB and routing model without precipitation. The mismatch should then be reduced.

Technical comments and clarifications

P2.L1: Clarify what you mean with "similar methods"

P2.L8: "did not quantify [it's effect] as we do here"

P3.L22: Rephrase. I think you are using one method (rating curve) to relate state and discharge, but three different methods to measure discharge.

P3. L29: Explain a bit more about how was the cross section area was determined. Particularly, why couldn't the area be determined for over 90% of the float discharge data.

P5. L10: Consider rewriting this. Pressure transduces may be installed through winter if protected from freezing with anti-freeze liquid.

P5.L26. Clarify if this is plus/minus 70 percent, or plus/minus 45 percent.

P5.27: Rephrase, this is unclear. Will you get back to these equations to revise them and update the river discharge dataset?

P5.L31-33: Confusing. Please rephrase.

P6.L4: Clarify how the model has been improved since van As et al. 2012. Some modifications are clear (i.e. MODIS albedo) while it is unclear if precipitation is a new modification or was part of the old model.

P7.L3: Rewrite equation so that it calculates albedo for each of the 100 elevation bins.

P7.L11: Explain where these in situ measurements originate from. Are they from the AWS stations or the K-transect?

P7. L24. Rewrite as this can be misunderstood. Meltwater run over the surface in ice sheet stream networks. However, Yang et al shows how these networks ends in moulins far from the margin and is routed subglacially from there.

P8. L30. Consider naming M, runoff to distinguish between meltwater production and runoff, given that not all meltwater reach the river.

P10. L15. Clarify what the Zo sensitivity test is.

P10. 25: Clarify what you mean with four "equal" portions.

P10.L 28: How does the accumulation rates play a role in the calculation of the p-value. Isn't the p-value just a function of elevation distribution?

P12. L16-27: The text about the development of englacial drainage system needs references or it should be made clear that those are speculations/hypotheses.

P12. L34. Figure 7 does not have monthly panels. . .

P13. L18. Clarify what the "long-term" ablation area refers to.

P13. L20. Clarify that these ice lenses are most likely to be in the higher elevation areas.

P14. L19. Rewrite. Some conclusions are indeed about the magnitude of discharge

P15. L20. This reads as if all the peaks are due to the combined effect of rainfall and melt. Is this true? It would be good to see the timing of the (modelled) rainfall events . P16.L8. This can't be seen in Figure 7. The delayed runoff agrees pretty well with the observations throughout the whole season.

Comments on figures

Figure 1: Show the entire catchment.

Figure 4. The x-axis shows 10 day smoothed temperature, right? Please clarify.

Figure 8: put the line about p=0 in the caption. Having it in the legend suggest that it is a line represented in the plot

Figure 9: Be consistent and use day of year or real dates, but not both

Reference

Carroll, D., Sutherland, D. A., Hudson, B., Moon, T., Catania, G. A., Shroyer, E. L., . . . van den Broeke, M. R. (2016). The impact of glacier geometry on meltwater plume structure and submarine melt in Greenland fjords. Geophysical Research Letters, 9739–9748. https://doi.org/10.1002/2016GL070170

---

## Author Comment (AC1) · 9 Apr 2017

X. Fettweis (Referee)

- The paper should more highlight that the considered catchment (in a very dry area) is likely not representative of other GrIS areas (for meltwater retention, lake drainage, . . .).

Indeed, the lower elevations of the Kangerlussuaq catchment are relatively dry in terms of precipitation compared to other ice sheet marginal areas. We do not consider it unrepresentative, but it is important to inform the reader on climatological particularities. Therefore we suggest the following: To add/change to the introduction: "Our study area is the relatively arid sector of the Greenland ice sheet east of the Kangerlussuaq settlement, located in southwest Greenland". We change a sentence in section 2.3 to

read: "... because of the arid climate that governs the lower elevations of the Kangerlussuaq catchment". In 4.3 we add: "Since supraglacial lakes are relatively abundant in the wide melt area of the Kangerlussuaq catchment, the impact of lake drainages on studies using our methodology would logically be smaller elsewhere in Greenland". Section 4.4 can read: "Climatologically though, Kangerlussuaq is arid in terms of precipitation due to blocking topography to the southwest (Van den Broeke et al., 2008; Johansson et al., 2015), providing this study with the possibility to study routing delays in an environment where complications by rain are minimal". We rewrite in section 4.6: "Also, (increases in) meltwater storage in supraglacial lakes (Fitzpatrick et al., 2014) are not calculated by the model". To the conclusions we add: "... takes 5-6 days to be released from this relatively arid sector of the ice sheet". We hope that these changes suffice in characterizing the Kangerlussuaq sector of the ice sheet to the reader.

- An interesting sensitivity experiment to evaluate the retention in firn (Section 4.6) should be to increase the winter snowfall by a factor 2. While the agreement is very good with obs, higher winter accumulation and higher melt could give similar results. Therefore, it is for me a bit too early to claim that there is not meltwater retention in this catchment. This should be confirmed by sensitivity experiments (e.g. Snowfall x 2 + Melt x 1.5).

Note that we do not make statements about the retention of meltwater in firn in the manuscript. Such retention is small compared to the catchment-total runoff even during peak melt events (Machguth et al., 2016). We would not be able to make solid claims on the topic even is meltwater retention in firn was twice as high, because this signal would drown in the uncertainties of observed river discharge and modelled ice sheet runoff. Instead, in section 4.6 we first mention that the model underestimates meltwater retention in winter-accumulated snow. Also we mention that we find no evidence for meltwater storage in the en- and subglacial environments. So in neither occasions do we mentioning retention in firn, in the supraglacial environment. Although your suggestion for a sensitivity experiment is an interesting one, it would not affect the

outcome of the study, and it will divert the attention of the reader away from the main conclusions. Therefore we suggest not to include this sensitivity experiment.

- Daily MAR outputs could be used in addition to check that melting routing delay used here are not too model dependent because it is very likely that the melting routing delays used here (Fig9) could compensate biases in the model. MAR has also its own but different biases. Evaluating how the routing delay is model dependent will add robustness in the paper. I can provide 7.5km daily outputs to the authors if they find that it is an interesting addition to their paper.

This is indeed an interesting suggestion and something to discuss for a follow-up study. As it stands, the current study is entirely based on observations for reasons of accuracy in terms of absolute values and temporal variability. We invested much effort into obtaining the most accurate as possible time series of ice sheet runoff to perform the routing delay calculation, because lower accuracy undoubtedly reduces the correlation with river discharge. Arguably, the ice sheet runoff values that we obtained at hourly resolution cannot be improved upon by a model that contains as accurate surface energy balance calculations, but is forced by observations at the lateral boundaries hundreds of kilometres away. Our forcing parameters, e.g. temperature, humidity, wind speed, solar/terrestrial radiation, and albedo, are all derived at the actual interface between the atmosphere and ice sheet, and should logically lead to an optimal result in terms of SMB calculations. Still, your idea is intriguing and deserves further consideration. Besides, recently I've learned that a study dealing with ice sheet routing delays is already underway, making use of MAR data as you suggest, providing an excellent opportunity to compare results.

- What does "ca." mean? It is used several times in the paper.

We use it as the abbreviation of "circa", which could also be "c.". If the editor wishes, we can use something else.

---

## Author Comment (AC2) · 9 Apr 2017

- 1. Regarding the rating curves (section 2.1). A more in-depth discussion about the previous ratings curves is needed. Were they created with some of the 90% of the float discharge measurements that now are discarded? Explain why they are so different.

In the paragraph that is dedicated to explaining these differences between the rating curves in section 2.1, we suggest to change/add the following: "The latter two differ from each other because of a revision in the cross-sectional area and larger availability of float-derived measurements at high stage – Hasholt et al. (2013) used ca. 140 float measurements to construct a rating curve. Yet because of the persisting problem of measuring channel depth during intermediate and high discharge, we have to reject

the (majority of the) float-derived discharge values for our study unless simultaneously the cross section could be determined. The further increase that our rating curve yields, we therefore speculate to be due to remaining uncertainties in cross-sectional area in Hasholt et al. (2013), and/or uncertainties in deriving the channel average velocity from surface float measurements in the rapid, supercritical flow through the irregular and seasonally heavily sedimented channel 1 (Fig. 2)." We feel that this describes in more detail on what grounds most float measurements were rejected, while also mentioning that it is uncertain that float measurements can be used to estimate channel-average flow velocities in this turbulent part of the river in the first place (see lower panel in Fig. 2 to get an impression).

- 2. Clarify what uncertainties are considered in the uncertainty estimates made with each the three discharge methods.

We suggest changing/adding the following: 1) "The depth- and width-integrated water flux has an uncertainty <10% combining sensor error, and bank-to-bank and depth integration of the point measurement, when the full width of the river is surveyed". 2) "Hasholt et al. (2013) estimate the float method to be accurate within 15%, representing the combined uncertainties in cross-sectional area, surface velocity determination, and calculation of depth-average velocity, although below we argue this to be an underestimate". 3) "The combined uncertainty of each depth- and width-integrated river discharge value, including sensor error in sediment-loaded water, is estimated to be . . .".

- 3. Regarding the gap filling method (section 2.2). Consider referring to the positive degree-day melt model as a motivation for using temperature for gap filling. Also, please provide the correlation/coefficient of determination and/or RMSE for fit between the observations and the temperature-based model.

We suggest rewriting the relevant sentence in section 2.2 to: "Whereas locally melting can be approximated by a linear temperature-index model, for the whole catchment

draining in Watson River we find that a power law approximates the relation between river discharge and air temperature (Fig. 4)", followed directly by Eq. 2. As to the second point, we can change the text to read: "FT equals 0.31 during the peak and late melt season (July and after, when the ice sheet ablation area has little to no snow cover), yielding a correlation of r = 0.74. FT is smaller (0.17) during the first half of the year (r = 0.65). The correlations are good considering that temperature cannot serve as a proxy for rain, jökullaups and other factors governing river discharge variability." The RMSE value of the fit is a less useful statistic when the fit error increases (substantially) with temperature, and therefore we argue to stick to reporting correlation values.

- 4. Alternatively, forego the temperature based gapfilling method all together. The paper would be simplified if the SMB model output with runoff delays would be used for gap filling the river discharge time series rather than the temperature based model. The advantage of the SMB model is that it physically based. This hinges upon that the runoff delay can be developed without the gap-filled time series.

This is a good suggestion, yet we argue for keeping the gap filling (section), primarily since we return to the relation between temperature and discharge when quantifying hypsometric amplification later in the manuscript. Also, we feel that it is important to inform the reader on how much of the discharge was not captured during low-flow situations in which the divers were not submerged, or during discharge events outside the instrumented period each year. But rest assured, the delays are calculated excluding the periods with temperature-derived discharge – as we suggest to make clearer in section 2.5: "We achieve this by finding the highest correlation between (observed, not temperature-derived) river discharge and catchment-total runoff . . .".

- 5. Regarding testing the SMB model performance (section 2.3). It would be good to quantify the difference between model estimates and the in situ ablation and accumulating data.

A quantitative description of model vs measurement mismatch is given in the last paragraph of section 2.3. We suggest to elaborate by (re)writing: "Over the course of seven melt seasons, the accumulated SMB at any time is modelled within 1.5 m ice equivalent of the measured values in spite of 20-40 m elevation differences (Fig. 5). At the end of the model run, the accumulated model error at the lower and middle weather station site is negligible; at the upper site the model underestimate SMB by one meter of snow, i.e. half a meter in water equivalent. The model annually overestimates winter snow accumulation by 0-0.5 m ice equivalent at low elevation; Van den Broeke et al. (2008) suggest that accumulation does not get recorded by low-elevation weather stations because snow mostly collects in the depressions between the ∼5-10 m diameter ice hummocks. At high elevation, winter accumulation and summer ablation (melt) are overestimated by up to 1 m ice equivalent in some years, which partially cancel each other out in terms of SMB (Fig. 5)".

- 6. Specify the uncertainties in the catchment delineation method and discuss the implications on the results and conclusions. For example, consider work by Ben Hudson about how DEM uncertainty propagates to catchment delineation (see ref by Carroll et al.).

We suggest to add the following text to section 2.4: "Yet ice sheet catchment delineation remains a relatively uncertain undertaking, e.g. due to uncertainties in bedrock topographical maps (Carroll et al., 2016) that were minimized by Lindbäck et al. (2015) by constructing a detailed bedrock map from ice-penetrating radar measurements. To our advantage, the Kangerlussuaq catchment is relatively wide at 30+ km near the ice sheet margin (Fig. 1), making its area-total runoff less sensitive to boundary shifts. Also, the large majority of meltwater is generated at low elevation, where catchment delineation is least uncertain due to its proximity to the watershed at the ice sheet margin". Since the catchment delineation and its uncertainties are the topic of the Lindbäck et al. (2015) study, we propose not to repeat their methods in greater detail, also because the main conclusions of our study are not sensitive to catchment size.

- 7. It is unclear how science question 1 is examined in section 3.2. The section

appears to quantify the effect of the factor three inter-annual variability on hypsometric amplification, but does not appear to explain it. It is a difficult section to follow so it may be in there, but difficult to follow. Consider rewriting.

Thanks for pointing this out. We propose to firstly simplify the research question 1: "Can we explain the large variability in ice sheet meltwater release by quantifying hypsometric amplification?", as well as question 2, to make them clearer and structured similarly: "Can we explain the moderation of ice sheet meltwater release by quantifying routing delays in the supra-, en-, sub- and proglacial environments?" Also, we can rewrite the last paragraph of section 3.2 to read: "The Watson River discharge time series confirms the value of the hypsometric amplifier in the Kangerlussuaq catchment, and thus that variability in meltwater release from the ice sheet is disproportional to the variability in atmospheric forcing of melt. Yet the uncertainty . . .".

- 8. The analysis about rainfall events and their impact on ice sheet meltwater production presented in the discussion warrants some deeper analysis. First, provide a more comprehensive identification of these events and how often they coincide with rainfall events. Second, to prove your argument that the temporal mismatch between the red and black lines in Figure 7 can be explained by rainfall events, run the SMB and routing model without precipitation. The mismatch should then be reduced.

Concerning your first point: excellent idea. We suggest identifying rain events exceeding 1 mm over a period of 6 hours, recorded by DMI in Kangerlussuaq, by adding + symbols to Fig. 7. This gives direct visual confirmation on how some rain events are captured well by the model, and some are not, leading to a mismatch between the black and red lines (river discharge and ice sheet runoff, resp.). We attach a revised figure. We suggest adding to section 4.4: "Meteorological measurements in Kangerlussuaq (Fig. 7; Cappelen, 2016) and near the ice sheet margin (Johansson et al., 2015) confirm precipitation in these periods, with only the 2015 event not exceeding a precipitation rate of 1 mm in 6 hours in Kangerlussuaq (Fig. 7j), and with positive temperatures yielding the possibility of liquid precipitation over the ice sheet". And also:

"Figure 7 illustrates that during most precipitation event that generate a spike in ice sheet runoff, the model performs well in capturing these events without overestimating river discharge. A clear example of a heavy rain event that was modelled with accuracy occurred in late August 2011 as described in detail by Doyle et al. (2015), (Fig. 7f). Climatologically though, Kangerlussuaq is arid in terms of precipitation due to blocking topography to the southwest (Van den Broeke et al., 2008; Johansson et al., 2015), providing this study with the possibility to study routing delays in an environment where complications by rain are minimal". As to your second point, we argue that the model performs very well in most situations with rainfall and that it only overestimates rainfall rarely. Therefore a model run without precipitation would generally not improve matters. We trust that the suggested changes to the text make it clearer to the reader that a modelled runoff overestimate occurs only in rare events, and that the outcome of the study is not impacted by these occurrences.

- P2.L1: Clarify what you mean with "similar methods"

We suggest changing this to "the input-output method", mentioned in the previous paragraph.

- P2.L8: "did not quantify [it's effect] as we do here"

Good addition, thanks.

- P3.L22: Rephrase. I think you are using one method (rating curve) to relate state and discharge, but three different methods to measure discharge.

True. We suggest changing the sentence to: "For this purpose, we measure discharge in three different ways".

- P3. L29: Explain a bit more about how was the cross section area was determined. Particularly, why couldn't the area be determined for over 90% of the float discharge data.

We propose changing/adding: "Only few measurements pass this criterion because

during intermediate to high flow, depth probing by tethered weight or iron rod is impossible, whereas we know there to be depth variations in channel 1 (Fig. 2) in excess of 2 m, due to erosion and deposition of bed load (sediment and gravel) (Hasholt et al., 2013)".

- P5. L10: Consider rewriting this. Pressure transducers may be installed through winter if protected from freezing with anti-freeze liquid.

Watson River at the Kangerlussuaq bridge does not run dry in winter, but it freezes over when water level is low, leaving the transducer encased in solid ice. We do use an anti-freeze liquid to prevent minor frost damage during the melt season, but that offers no protection for winter conditions. We can attempt to make this clearer by adding: "Due to the risk of frost damage when encased in solid ice . . .".

- P5.L26. Clarify if this is plus/minus 70 percent, or plus/minus 45 percent.

We will clarify by changing this to $\pm70\%$.

- P5.27: Rephrase, this is unclear. Will you get back to these equations to revise them and update the river discharge dataset?

"We return to these equations later when we use them for quantifying ice sheet hypsometric amplification of meltwater release".

- P5.L31-33: Confusing. Please rephrase.

We propose to shorten and clarify: "Hasholt et al. (2013) estimated the river discharge to be 4-11% of the annual values during these data gaps in 2007-2010. We find a smaller contribution during data gaps largely due to the revised stage-discharge relation".

- P6.L4: Clarify how the model has been improved since van As et al. 2012. Some modifications are clear (i.e. MODIS albedo) while it is unclear if precipitation is a new modification or was part of the old model.

[Figure]

Good comment, thanks. Two major changes/improvements were made to the model. We can add to the text of section 2.3: "One of the two major improvements made to our model is that we calibrated MODIS albedo . . .". And also: "The second large improvement to the model is an increase in temporal resolution. While for Van As et al. (2012) the daily MODIS resolution was an argument for running the SMB model in daily time steps, we recognize the need to resolve the daily cycle in ice sheet runoff. Therefore the current model version runs at an hourly time interval with a fixed daily albedo. The two above-mentioned, important changes to the Van As et al. (2012) model impact the melt and runoff calculations mostly by increasing. The SMB model is not in any way tuned to match river discharge.

- P7.L3: Rewrite equation so that it calculates albedo for each of the 100 elevation bins.

Note that this equation gives the correction factor to calculate "true" albedo from MODIS albedo. It does not calculate albedo, which is still very much an observation by MODIS (though adjusted) used in this observation-based study. To clarify, we suggest changing the text in section 2.3 to: "This is needed as we find from linear regression that MODIS on average gives albedo values lower than those derived from in-situ observations for solar zenith angles below ca. 74° (mean bias of 0.043): $\alpha\_true = \alpha\_MODIS + 0.114 \cdot \cos \theta\_noon - 0.032$ (3) where $\alpha$ is albedo and $\theta$ is the solar zenith angle."

- P7.L11: Explain where these in situ measurements originate from. Are they from the AWS stations or the K-transect?

We can write: "To test the model's performance, we compared its calculations of ablation and accumulation with independent in-situ measurements from the three weather stations."

- P7. L24. Rewrite as this can be misunderstood. Meltwater runs over the surface in ice sheet stream networks. However, Yang et al shows how these networks ends in moulins far from the margin and is routed subglacially from there.

Indeed – thanks. We suggest changing this to: "This delineation method is superior to methods entirely dependent on surface slope (e.g. Van As et al., 2012) as moulins and crevasses are abundantly present far from the ice sheet margin in the Kangerlussuaq region, yielding subglacial meltwater routing over large distances (Yang et al., 2016)."

- P8. L30. Consider naming M, runoff to distinguish between meltwater production and runoff, given that not all meltwater reach the river.

True, but here melt is estimated through a temperature-index method. It strikes us as odd to call this runoff instead of melt. Instead we suggest adding: "This is a fair assumption since meltwater retention in firn is of second-order magnitude for catchment-total runoff in this catchment even during extreme melting (Machguth et al., 2016)."

- P10. L15. Clarify what the Zo sensitivity test is.

We can clarify be writing: "A sensitivity test in which we vary Z0 between values representative of the lowest elevation of the regional ice sheet margin indicates . . .".

- P10. 25: Clarify what you mean with four "equal" portions.

We suggest writing: "We divide the ice sheet in four portions of equal surface area, and find . . .".

- P10.L 28: How do the accumulation rates play a role in the calculation of the p-value. Isn't the p-value just a function of elevation distribution?

We mean to say that the shape of the ice sheet in east Greenland may be different due to differences in accumulation. We propose to remove accumulation from this sentence altogether: "This relatively high factor is likely caused by the eastern ice sheet being bordered by high mountains, forcing the ice sheet to converge into valley glaciers with a relatively small area at lower elevations, as opposed to the generally less irregular margin elsewhere in Greenland."

- P12. L16-27: The text about the development of englacial drainage system needs

references or it should be made clear that those are speculations/hypotheses.

Agreed. We suggest adding a reference, and several words/phrasings to clarify what is interpretation of results, and what is speculation: "...and an underdeveloped englacial drainage system (Chandler et al., 2013). In July, commonly the peak river discharge month (Table 1), routing delays and the spread therein are smaller as surface snow is largely melted and presumably the englacial drainage system develops rapidly in response to increases in water supply. The reduced delays are most relevant at the lower and mid elevation bands from which most meltwater originates. Our results suggest that development of the englacial drainage system can occur over the course of mere days; for instance in the first half of July 2012, the drainage system shifted from below-average efficiency (larger delays in Fig. 9a) to above-average (smaller delays in Fig. 9b). After the peak of the melt season, in August, Fig. 9 suggests that the englacial drainage system remains capable of efficiently routing the dropping water volumes given the fact that delays are typically similar to those in July (Fig. 9c). In September, routing delays increase, presumably as drainage channels close and hydraulic efficiency reduces, most notably at lower and mid elevation where hydraulic efficiency is rapidly lost as water supply diminishes (Fig. 9d)."

- P12. L34. Figure 7 does not have monthly panels...

Thanks – we'll change this to Fig. 9 as intended.

- P13. L18. Clarify what the "long-term" ablation area refers to.

We suggest replacing "long-term" by "multi-annual".

- P13. L20. Clarify that these ice lenses are most likely to be in the higher elevation areas.

No problem: "Therefore we consider the role of largely impermeable ice layers in the firn of the high-elevation accumulation area (Machguth et al., 2016) to be minor for peak river discharge and road dam washout (Mikkelsen et al., 2016).

- P14. L19. Rewrite. Some conclusions are indeed about the magnitude of discharge

True. We can change this to: "In any case, changing the rating curve and thus the amplitude of the discharge signal does not impact our primary conclusions on hypsometric amplification and routing moderation."

- P15. L20. This reads as if all the peaks are due to the combined effect of rainfall and melt. Is this true? It would be good to see the timing of the (modelled) rainfall events.

We suggest adding the word "can" to this sentence to avoid this interpretation – not all discharge peaks stem from rain events. For the timing of the rain events we changed Fig. 7 accordingly – see above.

- P16.L8. This can't be seen in Figure 7. The delayed runoff agrees pretty well with the observations throughout the whole season.

Thanks for finding this error. We meant to state 100, not 1000 m3 s-1. We now intend to change the sentence to: "Figure 7 illustrates that modelled ice sheet runoff exceeds the river discharge values by 100-200 m3 s−1 during springtime of all years except 2010 when accumulation during the preceding winter was well below average (Tedesco et al., 2011)."

- Figure 1: Show the entire catchment.

We considered doing so already for the original submission, but found that adding the missing half of the catchment results in too little detail about the comparatively narrow proglacial area, the position of the discharge station, and the visible ice sheet features in the ablation zone. As opposed to showing the entire catchment, we could instead refer to a recently submitted study by Hasholt et al., who do show the entire catchment.

- Figure 4. The x-axis shows 10 day smoothed temperature, right? Please clarify.

Correct – we will correct this omission by adding "ten-day smoothed" to the figure caption.

- Figure 8: put the line about p=0 in the caption. Having it in the legend suggest that it is a line represented in the plot.

Good idea, we will do as you suggest, and add to the caption: "Note that a linear slope yields p = 0."

- Figure 9: Be consistent and use day of year or real dates, but not both.

These are our considerations in removing one of the two date formats: 1) The panels illustrate the delays per calendar month, therefore it would be odd not to mention the month. 2) The interpretation of the figure benefits from a visual comparison with Fig. 7 that makes use of the day of year as horizontal axis unit – therefore we cannot omit the day-of-year notation. We agree that using calendar dates only would simplify Fig. 9, but argue that there are clear benefits of having both date formats included in the figure.

---

## Author Response (AR2)

Dear editor,

Please find the revised manuscript below, with changes tracked (figures excluded). Many thanks for your efforts. We made all the requested changes, and a few more.

The exception is the notation of correlation as $r^2$. Naturally I agree with the argumentation you provide, and it is easy to change all occurrences of r into $r^2$. However, in section 2.5 we describe that we use r (not $r^2$) in calculating routing delays, and that we average the best time-delay solutions in the range where ice sheet runoff and river discharge correlate within $[r_{max}-0.01, r_{max}]$. Changing this to $r^2$ takes many days of recalculation, while the result will be virtually identical. Therefore, I prefer to keep using r throughout the manuscript. Also generally I prefer it over $r^2$ due to its ability to show anticorrelation. If you wish to keep the $r^2$ notation nonetheless, I suggest we do so throughout, except in section 2.5.

All the best,
Dirk van As

[revised manuscript text omitted]